# The inclusion of blastomeres into the inner cell mass in early-stage human embryos depends on the sequence of cell cleavages during the fourth division

**Junko Otsuki** [1,2]*, **Toshiroh Iwasaki**[2], **Noritoshi Enatsu**[2], **Yuya Katada**[2], **Kohyu Furuhashi**[2], **Masahide Shiotani**[2]

**1** Assisted Reproductive Technology Center, Okayama University, Okayama, Japan, **2** Hanabusa Women's Clinic, Kobe, Hyogo, Japan

* otsuki.midori.junko@gmail.com

## Abstract

The fate of the ICM in humans is still unknown, due to the ethical difficulties surrounding experimentation in this field. In this study we have explored the existing time-lapse recording data of embryos in the early stages of development, taking advantage of the large refractile bodies (RBs) within blastomeres as cellular markers. Our study found that the cellular composition of the ICM in humans is largely determined at the time of the fourth division and blastomeres which cleave first to fourth, during the fourth division from 8 cells to 16 cells, have the potential to be incorporated in the ICM.

## Introduction

In mouse embryos, the fate of the inner cell mass (ICM) has been known to be determined during divisions that occur from 8–16 cells [1, 2]. The outer cells give rise mainly to trophectoderm (TE). In contrast, cells positioned inside the embryo give rise to ICM. Recent paper has reported that depleting Sox21 levels leads to upregulation of Cdx2 and trophectoderm fate in mice [3]. They also reported heterogeneous gene expression of Sox21 in 4-cell and 8 cell mouse embryos from the highest to the lowest in blastomeres and speculated that heterogeneous gene expression in 4-cell stage in mouse embryos contributes to cell-fate decisions. This was consistent with previous studies showing gene heterogeneity in 4-cell stage mouse embryos [4–6]. However, how this heterogeneity in gene expression patterns occur and the mechanism(s) initiating these cell-fate remain unknown.

Long time ago, Sutherland et al. performed time-lapse observation and found that ICM is not associated with lineage from second and third division, but has a slight association with fourth cleavage division order in mouse embryos [7]. However, there is no information on the order of incorporation of blastomeres into the ICM in human embryos and cell-fate decisions.

In human oocytes, several types of morphological abnormality, such as smooth endoplasmic reticulum clusters and refractile bodies (RBs) are frequently observed. RBs consist of a mixture of lipids and dense granular material and exhibit yellow autofluorescence [8].

**Funding:** The authors received no specific funding for this work.

**Competing interests:** The authors have declared that no competing interests exist.

Only 1 or 2 large refractile bodies (> 5µm in diameter) are present in low proportion oocytes, while several small refractile bodies are commonly seen in almost all patients' oocytes. They are only formed in oocytes and never develop in the cells of an embryo during development from 1 cell to the blastocyst stage. While another dysmorphic phenotype, smooth endoplasmic reticulum clusters, disappears after fertilization, RBs remain present and almost unchanged in size at least until embryos reach the blastocyst stage. Although our recent study found that oocytes with RBs have a slightly lower potential for blastocyst development and implantation [9], more than 20% of high quality blastocysts with large RBs can result in healthy births and therefore these embryos can be good models for embryo development. Thus, our aim was to examine these early developmental stages using time-lapse recorded data, taking advantage of the large RBs within blastomeres as cellular markers.

## Materials and methods

This is a retrospective observational study based on existing data obtained during ART (assisted reproductive technology) treatment for patients. Time lapse recordings of embryo development are a regular part of the treatment and we merely took advantage of these pre-existing recordings. The informed consent regarding the use of this data was obtained from all patients concerned. This study was approved by the Institutional Review Board (IRB) of Hanabusa Women's Clinic (approval number: 2019–02).

### Time-lapse monitoring

Embryos which had large RBs larger than 5µm in diameter observed by a time-lapse system (EmbryoScope and EmbryoScope plus, Vitrolife, Japan) from June 2013 to December 2018 were reviewed. The time lapse images were captured automatically every 15 minutes, at 7 focal planes, which allowed us to track the position of the RBs in the embryos. A total of 201 large RBs in 143 fertilized oocytes progressing through normal 2-cell to 8-cell stages, which were from 111 patients, were traced until they developed into a blastocyst.

### Statistical analyses

Cluster analysis was conducted to group the blastomeres according to the order of cell division. Simple and multiple logistic regression analysis were both used to estimate the order in which the cells divided from the second to the fourth division, with the attainment of ICM defined as the endpoint. A chi-square test was also performed where appropriate. All statistical analyses were performed with EXCEL (2016, Microsoft) and EZR (Saitama Medical Center, Jichi Medical University, Saitama, Japan), which is a graphical user interface for R (The R Foundation for statistical Computing). Differences were considered statistically significant when the *P*-value was < .05.

## Results

Following the second division, from 2 cells to 4 cells, the rates of RBs that were distributed to the ICM of blastomeres which cleaved first and second were 20.0% (20/100) and 18.8% (19/101) respectively. During the third division from 4 cells to 8 cells, the rates of RBs that were distributed to the ICM of blastomeres which cleaved first to fourth were 24.1% (13/54), 28.1% (16/57), 10.3% (4/39) and 11.8% (6/51) respectively. During the fourth division from 8 cells to 16 cells, the rates of RBs that were distributed to the ICM of blastomeres which cleaved first to eighth were 35.1% (13/37), 30.8% (8/26), 26.9% (7/26), 30.4% (7/23), 5.3% (1/19), 4.8% (1/21), 4.0% (1/25) and 0% (0/24) respectively. The first 50% of cleaved blastomeres during the fourth

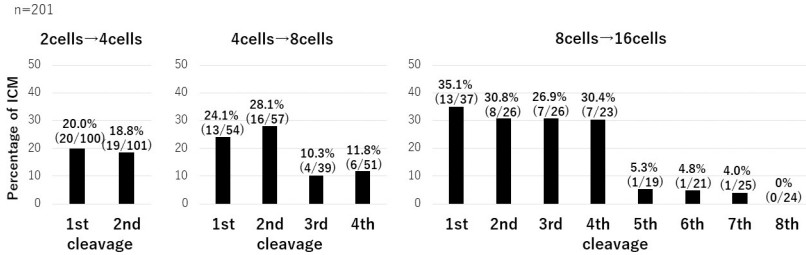

**Fig 1. The different rates of cells containing refractile bodies that were distributed to the ICM.** The relationship between the order of cell division and ICM allocation is shown. The first 50% of cleaved blastomeres during the fourth division had significantly higher rates of being incorporated in the ICM (p<0.001).

division had a significantly higher rate of being incorporated in the ICM (p<0.001) (Fig 1). The complete data is shown in S1 Table.

The rate of high quality blastocysts from embryos that had ICM from the RB(+) blastomeres which cleaved first to fourth was 92.9% (104/112). The rate of blastomeres which cleaved fifth to eighth was 89.9% (80/89). Among the embryos which were chosen for transfer, 30.6% (11/36) resulted in live births when they cleaved first to fourth, while 39.4% (13/33) resulted in live births when they cleaved fifth to eighth. There were no significant differences between the two order categories in the incidence of high quality blastocysts or live birth rates.

The order in which the cells cleaved during both the third and fourth divisions was identified by univariate logistic regression analysis as a significant predictor of which cells with RBs ultimately contributed to the ICM. However, multiple logistic regression analysis was only applied to the fourth cleavage. The third division was thereby eliminated as a confounding factor and the fourth division was found to be a predictor for ICM (OR:16, CI:4.1–63, p<0.001) (Table 1).

The dendrogram was obtained by hierarchical clustering analysis, which is based on the order of division, using the Ward's minimum variance method. The dendrogram revealed that samples can be divided into two clusters according to the order of cell division. The average order of the cells with RBs which cleaved during the third (4-8cell) and fourth (8–16 cells) divisions were 1.69 and 2.23 in cluster 1 (n = 107) and 3.28 and 6.41 in cluster 2 (n = 94) respectively. The ratio of the cells with RBs, which were allocated to the ICM/TE were 0.32 in cluster 1 and 0.05 in cluster 2 respectively (Fig 2).

Examples of the tracking RBs, which went to ICM and TE were shown in Fig 3 and Fig 4 respectively.

**Table 1. Univariate and multivariate logistic regression analyses with the attainment of ICM defined as the endpoint.**

| | Univariate logistic regression analysis | | | Multivariate logistic regression analysis | | |
|---|---|---|---|---|---|---|
| | OR | 95%Cl | P-value | OR | 95%Cl | P-value |
| The 2nd division | 0.93 | 0.46–1.9 | 0.83 | | | |
| (2-4cells) | | | | | | |
| (1st vs 2nd) | | | | | | |
| The 3rd division | 2.83 | 1.3–6.2 | <0.001 | 0.74 | 0.28–1.9 | 0.54 |
| (4-8cells) | | | | | | |
| (1st, 2nd vs 3rd, 4th) | | | | | | |
| The 4th division | 13.2 | 3.9–45 | <0.001 | 16 | 4.1–63 | <0.001 |
| (8-16cells) | | | | | | |
| (1st-4th vs 5th-8th) | | | | | | |

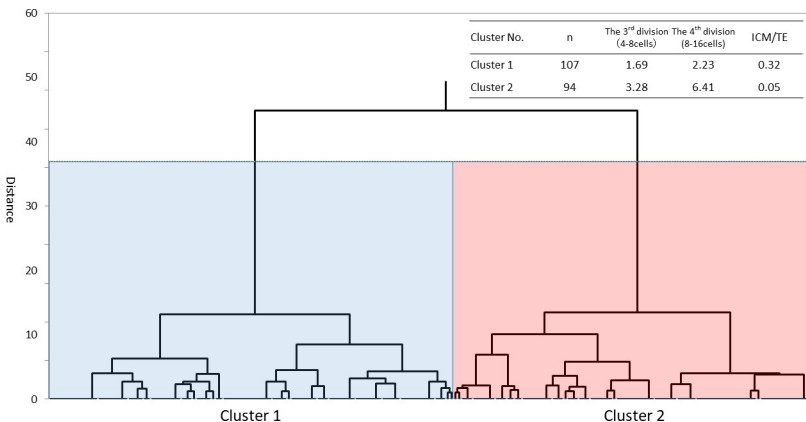

| Cluster No. | n | The 3rd division (4-8cells) | The 4th division (8-16cells) | ICM/TE |
|---|---|---|---|---|
| Cluster 1 | 107 | 1.69 | 2.23 | 0.32 |
| Cluster 2 | 94 | 3.28 | 6.41 | 0.05 |

**Fig 2. Dendrogram and the average order in which the cells cleaved during the 3rd and 4th divisions in each cluster.** The dendrogram revealed that samples can be divided into two clusters, according to the order of cell division. Cluster analysis showed that blastomeres which cleaved earlier tended to reach the ICM.

## Discussion

This study found that the cellular composition of the ICM in human embryos is largely determined at the time of the fourth division. Moreover, it was shown that blastomeres which cleaved first to fourth, during the fourth division from 8 cells to 16 cells, had a higher chance of ICM allocation. As there was no significant difference regarding the incidence of high quality blastocysts nor birth rates between embryos with blastomeres which cleaved first to fourth and those cleaved fifth to eighth, it can be considered that the presence of RBs themselves did not hinder the results in this study.

In mouse embryos, the majority of blastomeres appearing at the 16-cell stage, formed from the 8-cell stage, are reported to be asymmetrically divided [2, 10, 11]. They are polar-apolar pairs or polar-polar pairs [1, 12–14]. However, in contrast to this, Samarage et al [15] found that asymmetric division was uncommon and unnecessary for positioning the first inner cells of the embryo. They also reported that the occurrence of asymmetric divisions in 2-Dimensions (2D) was found to be more than 2 times higher than that in 3-Dimensions (3D).

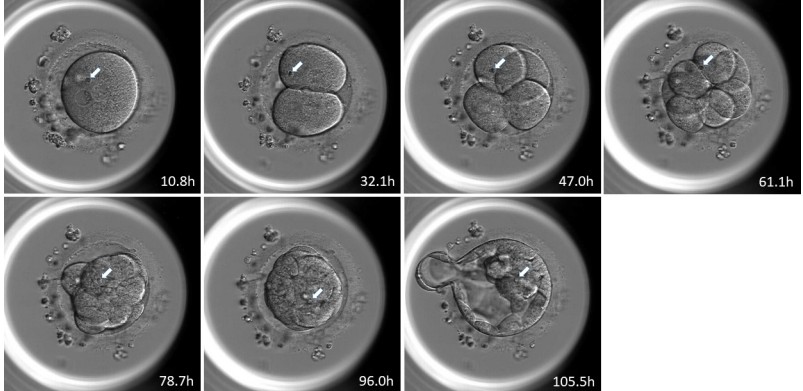

**Fig 3. Example of blastomere distribution to the ICM or TE obtained by tracking the position of refractile bodies.** In these sequential photographs, the refractile bodies were distributed to the ICM (Fig 3) or the TE (Fig 4). The white arrows indicate the large refractile bodies in 2PN zygotes and 2-cell, 4-cell, 8-cell, morula, and blastocyst stage embryos. The numbers indicate the time in hours from ICSI.

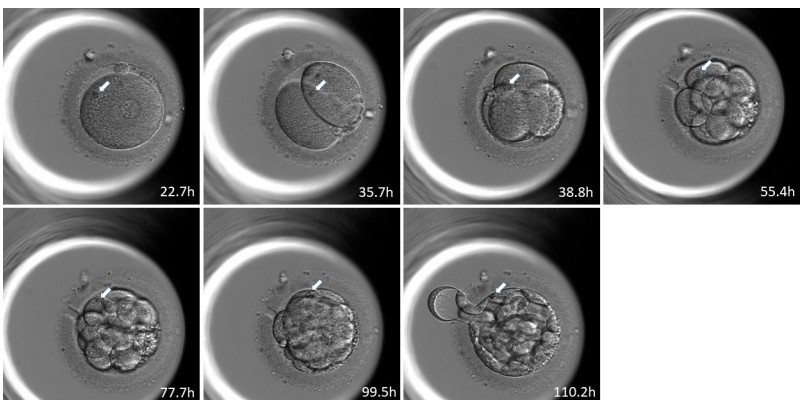

**Fig 4. Example of blastomere distribution to the ICM or TE obtained by tracking the position of refractile bodies.** In these sequential photographs, the refractile bodies were distributed to the ICM (Fig 3) or the TE (Fig 4). The white arrows indicate the large refractile bodies in 2PN zygotes and 2-cell, 4-cell, 8-cell, morula, and blastocyst stage embryos. The numbers indicate the time in hours from ICSI.

The polar cells are found to be in peripheral positions and mainly give rise to TE, whereas apolar cells are centrally positioned and give rise to ICM. Interestingly, transcriptome analysis of the isolated blastomers in mouse embryos at or before the 32-cell stage showed no significant variation between TE and ICM [16, 17]. In 5- to 8-cell stage human embryos, global single-cell cDNA amplification microarray analysis found that blastomere fate was not committed to ICM or TE [18]. This could be due to a loss of gene expression in isolated blastomeres as opposed to the hetelogeneous gene expression appearing in non-isolated embryonic blastomeres. In fact, comparing Oct4 and Cdx2 expression levels between intact 32-cell stage embryos and isolated blastomeres from 32-cell stage embryos revealed that cell-cell interaction is required to maintain gene expression levels in embryonic blastomeres [17]. Therefore, time lapse observation of blastomeres with RBs can accurately provide necessary data. The resolution of time-lapse observation systems for human embryo cultures has improved, making it possible to track large RBs as cellular markers in human oocytes. This study shows for the first time that the cellular composition of the ICM is largely determined at the time of the fourth division. However, the use of time-lapse observation systems continues to have certain limitations. Due to ethical issues with using human oocytes and embryos for experiment such as immunostaining and transcriptome analysis, there has been no such experimental data to study the fate of the ICM and TE in human embryos.

Recently, Yi et al. reported that only a subset of the blastomeres of 8-cell mouse and human embryos express keratins and that this biases their fate during trophectoderm differentiation [19]. As keratins stabilize the cortex, which may prevent cell internalization and also delay cell division, blastomeres devoid of keratin expression are able to cleave earlier. Thus, during the fourth division, from 8 cells to 16 cells, blastomeres which have cleaved first to fourth have the potential to be incorporated in the ICM. Noninvasive methods such as 3D time lapse observation systems are required to detect the exact position of blastomeres and will have the capacity to reveal the fate of ICM and TE in human embryos in the future.

## Supporting information

**S1 Table. Data set of cleavage division order and cell fate.** The first, second and third columns indicate the cleavage division order of blastomeres with refractile bodies during the 2–4

cell division, 4–8 cell division and 8–16 cell division.
(XLSX)

## Author Contributions

**Conceptualization:** Junko Otsuki.

**Data curation:** Junko Otsuki, Yuya Katada, Kohyu Furuhashi.

**Formal analysis:** Junko Otsuki, Noritoshi Enatsu.

**Investigation:** Junko Otsuki.

**Methodology:** Junko Otsuki.

**Project administration:** Junko Otsuki.

**Supervision:** Toshiroh Iwasaki, Masahide Shiotani.

**Validation:** Junko Otsuki.

**Visualization:** Junko Otsuki.

**Writing – original draft:** Junko Otsuki.

**Writing – review & editing:** Junko Otsuki.

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
