## [Decision Letter · Decision Letter 0]

3 Sep 2020

PONE-D-20-14274

The inclusion of blastomeres into the inner cell mass in early-stage human embryos depends on the sequence of cell cleavages during the fourth division

PLOS ONE

Dear Dr. Otsuki,

Thank you for submitting your manuscript to PLOS ONE. After careful consideration, we feel that it has merit but does not fully meet PLOS ONE’s publication criteria as it currently stands. Therefore, we invite you to submit a revised version of the manuscript that addresses the points raised during the review process.

We look forward to receiving your revised manuscript.

Kind regards,

Qiang Wu, Ph.D

Academic Editor

PLOS ONE

Journal Requirements:

2. Thank you for including your funding statemennt; "No"

Reviewers' comments:

Reviewer's Responses to Questions

**Comments to the Author**

1. Is the manuscript technically sound, and do the data support the conclusions?

Reviewer #1: Yes

2. Has the statistical analysis been performed appropriately and rigorously? 

Reviewer #1: Yes

3. Have the authors made all data underlying the findings in their manuscript fully available?

Reviewer #1: Yes

4. Is the manuscript presented in an intelligible fashion and written in standard English?

Reviewer #1: Yes

5. Review Comments to the Author

Reviewer #1: In this observational study, authors used large refractile bodies (RBs) within blastomeres as cellular markers to follow fate of the cells within preimplantation human embryos. They found that the inclusion of blastomeres into the inner cell mass in early-stage human embryos depends on the sequence of cell cleavages during the fourth division; the blastomeres which cleave first to fourth, during the fourth division from 8 cells to 16 cells, have a higher potential to be incorporated in the ICM. The conclusion is based on an analysis of 143 time-lapse recordings from 111 patients.

The observation is interesting, however, the mechanism or biological relevance remained elusive.

Can authors comment on blastocyst quality and pregnancy outcome from embryos that had ICM from the blastomeres which cleaved first to fourth vs. the blastomeres which cleaved fifth to eight?

6. PLOS authors have the option to publish the peer review history of their article (what does this mean?). If published, this will include your full peer review and any attached files.

Reviewer #1: No

---

## [Author Response · Author response to Decision Letter 0]

3 Oct 2020

Response to Reviewer 1

We appreciate the positive feedback and comments.

Comment 1: The observation is interesting, however, the mechanism or biological relevance remained elusive.

Answer to comment 1:

As a very recent paper published from Nature provided significant insight into our findings, we discussed the possible mechanism. This mechanism can be considered in the context in the light of both the Nature publication and our findings in this manuscript. The mechanism is explained in the revised manuscript (Page11-12 Line189-195 ).

Comment 2: Can authors comment on blastocyst quality and pregnancy outcome from embryos that had ICM from the blastomeres which cleaved first to fourth vs. the blastomeres which cleaved fifth to eight?

Answer to the comment 2: 

We have calculated the blastocyst quality and pregnancy outcomes of embryos that had ICM from the blastomeres which cleaved first to fourth vs. the blastomeres which cleaved fifth to eighth. This data has been explained and discussed in the revised manuscript (Page7 Line 116-122, Page10 Line159-162 ).

---

## [Editor Report · Decision Letter 1]

6 Oct 2020

The inclusion of blastomeres into the inner cell mass in early-stage human embryos depends on the sequence of cell cleavages during the fourth division

PONE-D-20-14274R1

Dear Dr. Otsuki,

We’re pleased to inform you that your manuscript has been judged scientifically suitable for publication and will be formally accepted for publication once it meets all outstanding technical requirements.

Kind regards,

Qiang Wu, Ph.D

Academic Editor

PLOS ONE
---

## [Editor Report · Acceptance letter]

8 Oct 2020

PONE-D-20-14274R1 

The inclusion of blastomeres into the inner cell mass in early-stage human embryos depends on the sequence of cell cleavages during the fourth division 

Dear Dr. Otsuki:

I'm pleased to inform you that your manuscript has been deemed suitable for publication in PLOS ONE. Congratulations! Your manuscript is now with our production department. 

Kind regards, 

on behalf of

Dr. Qiang Wu 

Academic Editor

PLOS ONE